# Mental health service delivery among adolescent girls and young women (AGYW) seeking HIV prevention and treatment services in central Kenya: A qualitative study of AGYW and healthcare providers' perceptions

Emmah Owidi[1], Kenneth Ngure[2,3], Peter Mogere[1], John Njoroge[1], Snaidah Ayub[1], Roy Njiru[1], Boaz Kipkorir[1], Edinah Casmir[4], Catherine Kiptinness[1], Tessa Concepcion[3], Tamara Owens[5], Pamela Kohler[3,6], Bradley H. Wagenaar[3,7], Shannon Dorsey[8], Pamela Y. Collins[9], Nelly Mugo[3,4], Jennifer Velloza[10]*

1 Partners in Health Research and Development, Center for Clinical Research, Kenya Medical Research Institute, Nairobi, Kenya, 2 School of Public Health, Jomo Kenyatta University of Agriculture and Technology, Nairobi, Kenya, 3 Department of Global Health, University of Washington, Seattle, Washington, United States of America, 4 Center for Clinical Research, Kenya Medical Research Institute, Nairobi, Kenya, 5 Simulation and Clinical Skills Center, Howard University, Washington, District of Columbia, United States of America, 6 Department of Child, Family, and Population Health Nursing, Seattle, Washington, United States of America, 7 Department of Epidemiology, University of Washington, Seattle, Washington, United States of America, 8 Department of Psychology, University of Washington, Seattle, Washington, United States of America, 9 Johns Hopkins Bloomberg School of Public Health, Baltimore, Maryland, United States of America, 10 Department of Epidemiology and Biostatistics, University of California, San Francisco (UCSF), San Francisco, California, United States of America

* jennifer.velloza@ucsf.edu

## Abstract

### Introduction

Common mental disorders (CMDs) are prevalent among adolescent girls and young women (AGYW) in high HIV-burden settings. However, mental health is underprioritized within HIV interventions targeting AGYW. We conducted a qualitative study to explore AGYW and healthcare providers' perceptions of mental health service delivery within HIV clinics.

### Methods

Between 16th February and 14th June 2021, we conducted in-depth interviews with AGYW receiving HIV services and healthcare providers from eight clinics in Central Kenya. Eligible AGYW were aged 16–25 years and reported mild-to-moderate CMD symptoms, determined by the Self-Reporting Questionnaire 20-item (SRQ-20) mental health screening tool. Eligible providers currently provided HIV or mental health services. Interviews explored AGYW's experiences with CMDs and factors influencing mental health service delivery by providers within HIV clinics. We analyzed data

**Data availability statement:** "The semi-structured interview guides and codebooks used to collect and analyze the qualitative interviews are provided as supporting information files. De-identified transcripts may be available upon consultation with the University of Washington and Kenya Medical Research Institute institutional review boards, in accordance with the ethical approval granted for this study. Full de-identified transcripts will be made available upon request and under appropriate data-sharing agreements to individuals submitting a methodologically sound proposal. Requests can be directed to the International Clinical Research Center at the University of Washington (Email: icrc@uw.edu) or the senior author (Email: jennifer.velloza@ucsf.edu)."

**Funding:** "This study was funded by a pilot award from the University of Washington Global Mental Health Program to the senior author (JV). The funders had no role in the study design, data collection and analysis, decision to publish, or preparation of the manuscript."

**Competing interests:** The authors have declared that no competing interests exist.

deductively and inductively using thematic analysis and organized findings using the socio-ecological model.

## Results

Median age among AGYW (n = 20) was 21 years (IQR:18–24), and SRQ-20 screening score was 9 (IQR: 8–11). Providers (n = 10) comprised seven females; and included six HIV and four mental healthcare providers. AGYW described experiences of CMDs due to multi-level risk factors, including HIV stigma, financial problems, and relationship challenges. AGYW reported a high demand for mental health services but described a systemic lack of access. Convenience and positive experiences with providers facilitated AGYW's access to services. Conversely, HIV care providers felt less confident in delivering mental health services due to inadequate mental health training compared to mental healthcare providers. Providers also reported inadequate training, poor referral systems, and unclear guidelines that hindered service delivery. AGYW and providers endorsed mental health service integration within HIV clinics to potentially reduce referral burden for AGYW and improve service quality.

## Conclusions

Our findings highlight gaps in mental health service delivery among AGYW receiving HIV services. Integrated service delivery within HIV clinics could improve AGYW's access to mental health services.

## Introduction

Adolescent girls and young women (AGYW) face a disproportionate burden of HIV and co-occurring common mental disorders (CMDs) such as depression, anxiety, and post-traumatic stress [1–3]. AGYW in sub-Saharan Africa (SSA) are three times more likely to experience depression than boys [4]. In Kenya, a recent study among 2106 AGYW aged 12–19 years found that 13 percent had ongoing depression symptoms, and one in five had experienced physical or sexual violence in the past 12 months [5]. AGYW receiving HIV services, including those who could benefit from oral HIV pre-exposure prophylaxis (PrEP) for HIV prevention and those on HIV treatment, are particularly vulnerable to CMDs [6–8]. Studies in SSA have reported a CMD prevalence ranging between 30–50% among AGYW receiving HIV services, compared to 10–40% among the general population [9–12]. Numerous barriers hinder mental health service access and delivery among adolescents receiving HIV services in SSA, including stigma, shortage of trained mental healthcare providers, poor treatment and referral pathways, and a lack of routine screening for CMDs [13–15].

   CMDs undermine HIV outcomes among AGYW by contributing to low service uptake and retention, poor adherence leading to uncontrolled viral load and poor treatment outcomes, limited engagement and persistence in care, and increased sexual risk behaviors, exacerbating AGYW's risk of HIV acquisition and transmission

[13,14,16]. Mental health is a critical driver of the effectiveness of behavioral and biomedical HIV prevention and treatment interventions [7,14], but has been largely neglected among the most vulnerable populations of AGYW in SSA [13,16]. Despite an abundance of research in SSA focused on HIV in adolescents, mental health is still poorly understood and under-prioritized within HIV research and interventions targeting AGYW [15,17]. Integrating mental health services within HIV clinics could improve access to mental healthcare among vulnerable AGYW populations [8].

AGYW in Kenya and other SSA countries have overlapping vulnerabilities to HIV and CMDs due to multi-level risk factors across individual, interpersonal, community, and structural levels. Previous studies have identified key risk factors, including physical and sexual violence, intimate partner violence, unfavorable sexual power relations, financial constraints, and low access to sexual and reproductive health (SRH) services [18–20]. AGYW in SSA experience low access to mental health services despite sufficient evidence regarding the negative impacts of CMDs on their HIV risk and outcomes [3,21,22]. In Kenya, there is a large gap in mental health service access and delivery within primary healthcare facilities, particularly among adolescents, with low rates of screening and referrals for CMDs within HIV clinics [23–26]. Kenyan AGYW with CMDs typically access mental health services such as counseling, psychotherapy, and rehabilitation at outpatient and inpatient mental health clinics located in sub-county, county, and national-level referral facilities, directly or through referrals by healthcare providers, teachers, and caregivers [23,27].

The World Health Organization's HIV prevention, testing, and treatment guidelines recommend the integration of mental health within HIV services to ensure the delivery of comprehensive and consistent care [28,29]. Several studies have endorsed the integration of mental health into SRH and HIV services to address co-occurring CMDs among people living with, affected by, or at risk of HIV [17,30–32]. Integration has been demonstrated to be feasible; and to potentially yield positive health outcomes such as viral suppression, reduced HIV acquisitions and transmissions, and improved physical and mental health among vulnerable populations such as adolescents receiving HIV services [7–11,29]. Recent studies in SSA have demonstrated the potential effectiveness of interventions supporting mental health integration for AGYW receiving HIV services [26,33]. Studies of mental health service integration among AGYW in Kenya have largely focused on pregnant and post-partum AGYW and those seeking SRH services [1,2,34,35]. Despite the increasing global and national policies supporting mental health integration within HIV services, there is limited data on mental health service delivery and access among AGYW receiving HIV services, particularly in Kenya. We conducted a qualitative study to understand the perceptions around potential expansion of mental health service access and delivery within HIV clinics among AGYW receiving HIV services and healthcare providers in Central Kenya. Our findings could inform mental health service integration within HIV clinics in Kenya and similar settings.

## Methods

### Study design

This was a formative qualitative study conducted as part of the larger JiTunze pilot study (funded by the University of Washington Global Mental Health Program). This qualitative study aimed to inform the development of a simulated patient encounters (SPE) implementation strategy applied for training HIV care providers to improve mental health service delivery within Kenyan HIV clinics. The design and findings of the SPE implementation strategy have been previously described [26].

### Study setting and population

We conducted the study in Thika, an urban center in Kiambu county in Central Kenya, located approximately 40 km from the capital, Nairobi, with a population-level HIV prevalence of 2.7% and a large surrounding peri-urban and rural population [36]. Kiambu county hosts approximately 250,000 AGYW of ages 15–24, who account for over 1,000 new HIV infections annually [37]. The primary site for the study, Thika sub-County Hospital (Level 5), was ideal for the study as it provides HIV prevention and treatment services to a large population of AGYW aged 10–24 (233 living with HIV; and 9 on

PrEP). We also included seven surrounding healthcare facilities to broaden the diversity of the study population. These included Partners in Health and Research Development (PHRD), a clinical research site that implements HIV prevention and treatment trials among diverse populations including AGYW, Makongeni dispensary (Level 2), Kiandutu health centre (Level 3), Ruiru sub-County hospital (Level 4), Kiambu County referral hospital (Level 5), Mathari National referral hospital (Level 6), and Kenyatta National hospital (Level 6).

**Participant recruitment.** We recruited 20 AGYW and 10 healthcare providers for participation in qualitative in-depth interviews between 16th February and 14th June 2021. AGYW were recruited from Thika sub-County Hospital and the PHRD clinic. Healthcare providers were recruited from Thika sub-County Hospital and six surrounding facilities to capture broader perspectives on mental health service delivery to AGYW within the larger catchment area that provides referrals to the primary study site.

AGYW were referred for study-related screening and enrollment by HIV care providers from Thika sub-County Hospital and the PHRD clinic. Eligible AGYW were aged 16–25 years and were currently receiving HIV prevention, treatment, or testing services at either clinic at the time of enrollment. HIV care providers at the two facilities were sensitized about the study to support the recruitment of AGYW by identifying potential participants while providing HIV services. Once the providers identified AGYW who fit the eligibility criteria, they informed the AGYW about the study and referred those who expressed willingness for study participation to the study staff at the PHRD clinic (approximately 1.5 kilometers distance) for screening and enrollment. This assured AGYW of confidentiality, enhanced their comfort in study participation, and built trust between the AGYW and the study staff. Once the AGYW arrived at the PHRD clinic, they were screened for study participation by a study psychosocial counselor in a private clinic room using the Self Reporting Questionnaire 20-item (SRQ-20) mental health screening tool [38] (S1 File), to assess for elevated symptoms of CMDs. AGYW with mild-to-moderate CMD symptoms (determined by a score of 7–14 on the SRQ-20 screening tool), were eligible for in-depth interviews. AGYW with scores above 14, indicating severe CMD symptoms, were excluded from interviews due to potential impacts on the capacity to consent to study participation, and risk of harm related to emotional distress from discussing sensitive topics or traumatic experiences [39]. These AGYW were instead referred to a study clinical psychologist for psychosocial support.

We recruited providers from surrounding HIV and mental healthcare facilities. The providers were eligible if they were aged ≥18 years and had a current role in providing HIV or mental health services in an HIV, mental health, or primary healthcare facility located around Thika and Nairobi areas. The providers were recruited physically during in-person facility visits or using phone calls through which they were invited for study participation.

**Participant sampling.** We used stratified purposive sampling to capture varied experiences of mental health service delivery among different demographic groups [40]. We included AGYW in different age groups (16–18, 19–22, and 23–25). These AGYW were receiving a variety of HIV prevention and treatment services including: PrEP, HIV testing services (HTS), post-exposure prophylaxis (PEP), and antiretroviral therapy (ART).

We also purposively sampled HIV and mental healthcare providers to include different cadres (HIV counsellors, clinical officers, nurses, psychiatrists, and psychologists), healthcare facility representation, service delivery experience, and educational backgrounds (college diploma and university degree). The overall sample size of 20 AGYW and 10 providers was determined by thematic saturation, whereby we stopped interviews once no new data emerged [40].

## Data collection

All in-depth interviews with AGYW and providers were conducted between 16th February and 14th June 2021 by a trained female qualitative researcher (EO) in-person at the PHRD clinic for AGYW, and at the respective facilities or by phone for providers. Interviews were conducted in participants' preferred language (English or Kiswahili), were audio-recorded, and each lasted approximately one hour. Since the study took place during the COVID-19 pandemic, relevant infection prevention guidelines were observed, including hand sanitizing, mask-wearing, and social distancing during participant screening, recruitment, and data collection procedures.

We used semi-structured interview guides comprising of theme-based, open-ended questions with probes to clarify responses and facilitate the flow of the discussion. Among AGYW, interviews explored themes around experiences of CMDs, language used to describe CMD symptoms, and factors influencing access and utilization of mental health services (S2 File). Among providers, we explored factors influencing delivery of mental health screening, counseling, and referral; clinic culture and perceived competence around mental health service delivery; perception of Kenyan Ministry of Health mental health policies on HIV and mental health service integration, and training and supervision needs (S3 File). We paused the interviews if any AGYW showed severe symptoms of mental health distress until they felt comfortable enough to continue with the interviews. We referred these AGYW to a psychosocial counselor at the PHRD clinic for psychosocial support after the interviews were completed.

## Data management and analysis

We directly transcribed audio-recorded interviews verbatim regardless of language and translated Kiswahili transcripts into English. We analyzed interview data deductively and inductively using thematic content analysis [41], to identify AGYW and providers' views on factors influencing mental health service access and delivery within HIV clinics. All analyses were performed on de-identified data using Dedoose version 9.0.18 (SocioCultural Research Consultants, LLC, Los Angeles, USA) [42]. First, authors EO and JV reviewed all transcripts, generated key themes, and developed separate codebooks for AGYW (S4 File) and providers (S5 File) using *a priori* codes from the interview topic guides. The authors iteratively revised the codebooks based on emerging themes from the interview transcripts, then selected a subset (10%; n = 3) of AGYW and provider interviews and double-coded them to ensure consistency. The authors continuously reviewed and revised the codes and themes and resolved intercoder discrepancies over weekly meetings until an agreement was reached. The remaining transcripts were independently coded by EO and reviewed by JV. Key themes and illustrative quotes from the interviews were grouped throughout the coding process. Once coding was complete, the authors summarized key themes from AGYW and provider interviews in memos and compared findings between the two groups.

We used the socio-ecological model to categorize themes around the individual, interpersonal, community, and structural factors influencing mental health service delivery, access, and utilization among AGYW receiving HIV services and providers delivering HIV and mental health services in the respective healthcare facilities. The socio-ecological model, proposed by Bronfenbrenner in the late 1970s, described five levels of nested social systems that influenced individuals' development: micro-, meso-, exo-, macro-, and chronosystems [43]. This model has been adapted within public and mental health research to explain the multi-level factors influencing individuals' health behaviors and risk factors, which include: intrapersonal, interpersonal, organizational, community, policy, and societal factors [44,45]. The socio-ecological model was relevant for our study as it provided a comprehensive understanding of the factors influencing AGYW's mental health, due to the recognition that adolescents and young people's (AYP) social environments, including their individual experiences, family and peer relationships, and community contexts (e.g., school, and neighborhood) significantly influence their health behaviors and mental health outcomes [46,47].

We presented our findings of the key themes in the subsequent sections, organized within the different levels of the socio-ecological model (individual, interpersonal, community, provider, service, and structural-level factors). The key themes included: AGYW experiences and symptoms of CMDs, risk factors for CMDs among AGYW, experiences of mental health service delivery within HIV clinics, factors influencing mental health service access and delivery among AGYW, and benefits of integrating mental health services within HIV clinics.

## Ethical statement

The University of Washington and Kenya Medical Research Institute institutional review boards approved this study. All participants provided written informed consent before study participation by appending their signatures or thumbprints to the informed consent form and were given a copy of the consent documents for their records. AGYW aged below 18

years also provided individual informed consent since in Kenya, adolescents aged 12 years and older can consent for themselves without a waiver of parental permission for the purposes of research on mental health, HIV, and sexual and reproductive health [48]. Participants interviewed via phone provided verbal consent to study participation through an audio-recorded phone conversation in which they stated their names as proxy for written consent. All interview and consent audio recordings were stored in a password-protected computer in separate folders and then archived in a secured external hard drive pending deletion 3 years after completion of data analysis and study closure.

## Results

### Participant characteristics

The median age among AGYW was 21 years (interquartile range [IQR]: 18–24), and the SRQ-20 screening score was 9 (IQR: 8–11). The AGYW included those living with HIV (n = 7) and those without HIV (n = 13); with services received including HIV treatment (n = 7), PrEP (n = 7), and HIV testing and/or PEP (n = 6). The providers included six HIV and four mental healthcare providers (two HIV counselors, three clinical officers [physician assistant equivalent], three nurses, one psychologist, and one psychiatrist). Providers had a median age of 37 years (IQR: 30–44). The sample included 70% (n = 7) females, with a median of 7 years of experience (IQR: 5–17) in HIV or mental health service delivery; their educational backgrounds included diploma (n = 6), higher diploma (n = 2) and degree (n = 2). The providers represented seven HIV and mental healthcare facilities: Thika sub-County hospital (n = 4), and (n = 1) each from Makongeni dispensary, Kiandutu health centre, Ruiru sub-County hospital, Kiambu County hospital, Mathari National hospital, and Kenyatta National hospital.

### AGYW experiences and symptoms of CMDs

AGYW often described any experiences of CMDs as 'stress' and used words in English, Kiswahili, or 'Sheng' (an urban slang popular among Kenyan youth) languages to describe their mental health symptoms. AGYW used words like: 'niko na stress' (Sheng- I have stress), 'kufikiria sana' (Kiswahili- thinking a lot), 'kitu inanisumbua' (Kiswahili- something is disturbing me), 'mawazo mengi' (Kiswahili- many thoughts), 'sifeel poa' (Sheng- I don't feel fine), and 'siko sawa' (Kiswahili- I am not okay). AGYW also described CMDs using words reflecting self-value perceptions, like 'I feel useless' or 'I feel hopeless'. There were some differences in descriptions of CMD symptoms relative to AGYW's SRQ-20 screening scores. Symptoms like self-isolation, changes in sleep and appetite, and drug and alcohol use were more likely to be reported among AGYW with lower SRQ-20 screening scores (7–10). Headaches, self-harm, and suicide ideation were mostly reported among AGYW with higher SRQ-20 screening scores (10–12), as one AGYW seeking ART shared below:

> So that is when I started throwing away the drugs [HIV treatment or antiretroviral therapy (ART)]. I would go to the kitchen and see that it is only a knife that can help me. Even my own mother, I have never told her that I wanted to kill myself. It would just get to a point, and I would ask her, "Why should I be like this? Why can't I die and go away? (AGYW, Age 18, Seeking ART, SRQ-20 Screening Score 12).

### Risk factors for CMDs among AGYW

**Individual-level risk factors for CMDs.** Among AGYW living with HIV, HIV stigma was a common risk factor for experiencing CMDs. This included internalized stigma attributed to challenges coming to terms with their HIV status, fear of intimate relationships due to negative self-image related to their HIV-positive status, fear of rejection, and fear of potentially transmitting HIV to their sexual partners. Among AGYW without HIV, CMD symptoms were linked to financial constraints, especially among single-parenting adolescents, and SRH issues such as unintended pregnancy, post-abortion, and post-partum complications. Other issues included infertility, sexually transmitted infections (STIs), and challenges negotiating condom use and HIV testing with sexual partners, as one AGYW seeking PrEP reported below:

The time I was feeling stressed was when I got pregnant and the young man that I loved left, and he said that he did not even want the pregnancy…I was so stressed, and I was asking myself why." (AGYW, Age 20, Seeking PrEP, SRQ-20 Screening Score 10).

**Interpersonal and community-level risk factors for CMDs.** Among AGYW living with HIV, CMD symptoms were also attributed to experiences of HIV-related stigma from discrimination, abuse, and withholding of emotional, material, and treatment support by peers, partners, and caregivers due to AGYW's HIV status. Furthermore, perceived and anticipated HIV-related stigma was related to lack of privacy in HIV clinics, schools, and the community, which presented a risk of inadvertent HIV status disclosure and affected discreetness in taking medication. These factors contributed to mental health distress among AGYW living with HIV, as one AGYW described her stressful experience of taking ART medication while in school:

After being told [about my HIV status] I was very stressed, because you would wonder, you have gone to high school, and it is a boarding school, and you are taking those drugs. It becomes a challenge…it is only you using the drugs, and for those drugs, it is a must…So, it got to a point that students started asking me, "Which problem is this—every morning and you have to keep a certain time!" (AGYW, Age 18, Seeking ART, SRQ-20 Screening Score 12).

Among AGYW without HIV, mental health symptoms were mostly related to relationship problems, including emotional and physical abuse, mistreatment and violence by partners and caregivers, conflicts and separations among parents, sexual violence, and pressure from peers and community members for AGYW to attain certain high life goals and lifestyles.

### Experiences of mental health service delivery within HIV clinics

**AGYW experiences of mental health service delivery.** Less than half of the AGYW reported having ever received mental health services within HIV clinics, and the majority did not know where to access the services if they needed them. However, we found that most AGYW who reported receiving mental health services in HIV clinics had only received HIV adherence counseling and follow-up specific to HIV and STI prevention and treatment services (e.g., HIV testing, HIV status disclosure, and medication adherence) from HIV counseling providers, as one AGYW seeking HTS and PEP described below:

Okay, what she [provider] did is the counseling: she told me about how I can protect myself, about how I can also use PrEP, she also told me about how I am going to take my medication (PEP). Yeah, she just told me about preventive measures; about how I can also cope with people who are HIV-positive [living with HIV] and not get infected. (AGYW, Age 24, Seeking HTS/PEP, SRQ-20 Screening Score 11).

A few AGYW had received mental health counseling related to gender-based violence, unintended pregnancy, relationship challenges with partners, and HIV stigma, from providers based at healthcare facilities within their communities. These AGYW described four broad strategies they used to access the mental health services: (1) approaching providers and asking for help; (2) referral from one provider to another; (3) providers approaching AGYW presenting with mental health symptoms; (4) and caregivers seeking support from providers on behalf of AGYW. However, within HIV clinics, AGYW typically received only the specific HIV services they sought from providers, and some were not offered any mental health services while receiving services like HIV testing or ART, despite needing it. AGYW said that receiving mental health support from HIV care providers would be beneficial for getting relief from mental health burdens, empowering them, and helping them potentially find solutions to their problems, as one AGYW seeking HTS and PEP described below:

I know the benefits of being counseled, I know you can get help, you cannot drown in your stress or in those mental issues. It helps you; you can be able to move on, [and] you can be able to accept…Being counseled is helping me accept that it already happened, and we can't undo it. (AGYW, Age 18, Seeking HTS/PEP, SRQ-20 Screening Score 7).

Despite their varied experiences with receiving mental health services, AGYW generally perceived mental health services positively, recognized their importance, and expressed a high demand, acceptability, and willingness to receive the services within HIV clinics.

**Provider experiences of mental health service delivery.** Among HIV care providers, experiences with mental health service delivery to AGYW were mostly related to addressing stress and anxiety symptoms. However, mental healthcare providers had experience with addressing more severe psychotic and depressive symptoms among AGYW. Generally, mental healthcare providers felt more comfortable, competent, and confident in delivering mental health services to AGYW than HIV care providers. HIV care providers' lower confidence in providing mental health services was mainly due to limited skills. Mental health services provided by HIV and mental healthcare providers included screening, mental health counseling, and referrals. Screening was primarily conducted by clinical assessment of clients' history and symptoms. However, more specific screening tools were also used for further CMD diagnosis. Among both HIV and mental healthcare providers, mental health counseling provided to AGYW was mostly related to addressing internalized HIV and mental health stigma, social issues like abuse, gender-based violence, and conflicts among AGYW and their caregivers or partners, SRH issues such as unintended pregnancies, and contraceptive use, substance use addiction, grief, and life skills. HIV and mental healthcare providers also provided referrals for AGYW to other specialists or facilities for further evaluation and diagnosis of CMDs, as well as linkages to support groups for mental health, medication adherence, and substance use addiction issues, as one mental healthcare provider narrated below:

> We also do support groups, just to help them [AGYW] have an appreciation that: one, they are not alone, they are quite a number, and two, as they share, they can also share different strategies and information about how best to cope with the different challenges that they are facing. So, it is very key in terms of provision of services at the youth center—I mean at the…'Facility X' HIV clinic. (Male Psychiatrist, HIV Clinic).

**Routine mental health screening and referrals within HIV clinics.** Routine mental health screening and referrals within HIV clinics were limited to AGYW living with HIV with high viral load or serious adherence issues. Screening tools used included CAGE-AID [49] and CRAFFT [50] for alcohol and drug abuse screening, Morisky Medication Adherence Scale (MMAS-4) [51] for adherence screening, and Patient Health Questionnaire-9 (PHQ-9) [52] for depressive symptoms screening. These tools were integrated into the Electronic Medical Records (EMR) system, for screening clients with high viral load, to rule out mental health or substance use issues as possible adherence barriers. HIV care providers provided referrals to AGYW if they felt unequipped to offer them mental health support, and if AGYW were uncooperative or preferred specific providers. Most AGYW referrals were done through 'referral by escort', whereby HIV care providers physically escorted AGYW to relevant providers, e.g., psychologists located within HIV clinics, or externally to mental health clinics located within the facility. This ensured AGYW reached referral points and enabled HIV care providers to introduce AGYW and brief the mental healthcare providers about their case, ensuring AGYW's comfort and cooperation to share their issues with providers, as one HIV care nurse reported below:

> Mostly, we encourage escorting them [AGYW] to those referral centers. If you think the client is having anxiety or depression, you just take her to the psychologist…but you have to escort [them]; otherwise, if you tell them to go, they won't go, because they won't be able to open up to everybody. (Female Nurse, HIV Clinic).

### Factors influencing mental health service access and delivery among AGYW

**AGYW individual and interpersonal-level factors.** Various factors influenced AGYW's decisions to access mental health services within HIV clinics including receiving services from familiar providers, attending few service points, short

waiting times, flexible clinic operating hours, and incurring no costs for services. Positive interpersonal experiences with providers, including perceived good quality reception, empathy, and competency, also motivated AGYW to seek mental health services. AGYW preferred providers they perceived as welcoming and friendly, used polite language, listened, showed genuine interest, reassured and encouraged them to talk, and were non-judgmental, as one AGYW seeking PrEP narrated below:

> You know, you could also enter the room, and you feel that that [provider] is not even a person to share your problems with, and you would rather even leave…There are those [providers] you usually find are in a bad mood all the time, and they usually don't even greet you. (AGYW, Age 23, Seeking PrEP, SRQ-20 Screening Score 9).

The key obstacles to AGYW accessing mental health services included preferences related to provider demographic characteristics such as age and gender while seeking services in HIV clinics, with most AGYW reporting being more comfortable with female providers. AGYW perceived younger providers as more likely to understand their issues, but mature providers as more empathetic, likely to maintain confidentiality, and more invested in their well-being. Other concerns among AGYW included provider confidentiality, embarrassment in sharing their issues with providers, especially during first clinic encounters, and fear of stigma and discrimination from being seen at the HIV clinics by people known to them, as one AGYW seeking HTS and PEP shared below:

> What may make me—I may feel embarrassed because people know that this is an HIV clinic. So, "She is going because maybe she has HIV". Yeah, the embarrassment that comes with it. (AGYW, Age 18, Seeking HTS/PEP, SRQ-20 Screening Score 7).

**Provider individual and interpersonal-level factors.** Overall, HIV and mental healthcare providers felt motivated to provide mental health services to AGYW, but various factors affected their ability and willingness to deliver the services. Among HIV care providers, inadequate mental health training (e.g., related to knowledge of screening tools and AGYW-friendly communication, including rapport-building and empathy skills) was a key barrier to mental health service delivery. Conversely, mental healthcare providers had adequate training and considered mental health services as their core evaluable role. Consequently, most HIV care providers preferred referring AGYW with CMDs to HIV clinic adherence teams—comprised of psychologists and HIV counselors—or other trained providers. HIV care providers also reported difficulties when counseling AGYW with persistently high viral loads, noting that these AGYW often became angry with providers, refused to answer adherence-related questions, and defaulted on their monthly clinic visits, as one provider described below:

> The challenge comes with the high viral load ones [AGYW] where you have now to see them every month and ask, "Why, why, why is your viral load up?" It gets to a point when they start getting angry at you. So, they get angry at you and then they start just keeping quiet at you, so it is you now to decide how you will probe and get to the bottom of the issues…So, like those ones now…I can refer very fast. (Male Clinical Officer, HIV Clinic).

Furthermore, some HIV clinic staff perceived AGYW as 'not easy to work with', 'troublesome', or 'difficult' clients that required a lot of provider time and patience. Others reported difficulty in understanding words and language AGYW used during counseling, while others perceived mental health services as 'extra work'.

Conversely, among mental healthcare providers, key challenges included difficulty of AGYW gaining insight (i.e., ability to recognize and accept their CMD symptoms), AGYW declining mental health services and medication, mental health stigma, and lack of treatment support from caregivers and partners. Mental healthcare providers also reported observing stigma and discrimination from other providers, especially those in non-psychiatric departments, who avoided interacting

with clients with mental health disorders and used stigmatizing words like '*Mwendawazimu*' (*Kiswahili*- mad person) to refer to clients, as one mental healthcare nurse expressed below:

> In a general outpatient like in a certain hospital, you find that there are those providers who do not want completely to be associated with those people with mental illnesses; they actually run away from them, they probably believe they might also be aggressive and violent and very harmful or dangerous people. (Male Nurse, Mental Health Clinic).

**Service and structural-level factors.** Among HIV care providers, structural factors affecting mental health service delivery to AGYW included a lack of funds for provider training and inadequate rooms for adolescent-friendly consultations, which resulted in a lack of privacy for AGYW. HIV care providers also experienced difficulties escorting AGYW to and from mental health clinics for their referrals, due to challenges around lack of time, busy clinic workloads, and staffing shortages. Further concerns included the lengthy time needed to provide HIV services to AGYW since providers had to complete more documentation for AGYW aged up to 25 years who were in HIV care compared to adults. Providers also reported concerns about competing roles, high workload, time constraints related to high client volumes within public HIV clinics, and a lack of clear guidelines and practical tools to support HIV care providers on integrating mental health screening into HIV service delivery. These factors contributed to providers missing out on mental health within routine HIV care, as one clinical provider highlighted below:

> I wish we had a—what should I call it? A defined tool that would guide you on specific, like the major ones [CMD symptoms] that we need to screen for…because most of the time, us clinicians miss [mental health screening]. You will miss [it] because maybe you forgot to ask, or maybe it is because there is no directive that you can ask directly. (Female Clinical Officer, HIV Clinic).

Among mental healthcare providers, poor referral pathways, which underprioritized AGYW referred from HIV clinics, resulted in long waiting times as AGYW had to queue for services twice—at HIV and mental health clinics. Furthermore, there was a lack of transportation for external referrals, unclear systems for tracking and follow-up of clients, and a lengthy time needed to provide screening and counseling to AGYW. Staffing shortages and high turnover also affected services, with the few available trained mental healthcare providers having to create time outside working hours to follow up clients. Limited resource allocation to mental health units within public health facilities, shortage of in-patient facilities, and restrictive facility policies further limited mental health services delivery to AGYW, as one mental healthcare provider described below:

> In [public referral facility X], there is no admission for children; we don't have a unit whereby we manage children under the age of 18 years. So, whoever is admitted has to be more than 18 years. But in [private facility Y], we do admit younger people, even children who suffer from mental illness. (Male Nurse, Mental Health Clinic).

### Benefits of integrating mental health services within HIV clinics

Both AGYW and providers supported the integration of mental health services within HIV clinics, acknowledging the need for these services among AGYW. Providers specifically emphasized the need for integration due to the high vulnerability of AGYW to CMDs, particularly among those receiving HIV treatment. This concern was attributed to the high numbers of AGYW who are currently referred to psychiatric clinics and those attending both HIV and psychiatric clinics simultaneously. However, both AGYW and providers reported a lack of integration of mental health services within HIV services for AGYW, as one HIV counselling provider reported below:

> If I can speak frankly, it [mental health] has not been tackled the way it should be. That has always been a gap. It's like we only feel like the need which brings them [AGYW] is what we are supposed to handle…but as time goes on,

especially the ones whom you find that their viral load has started multiplying, that is when we will realize there is a problem. (Female HIV Counselor, HIV Clinic).

Overall, integration was anticipated to result in fewer AGYW referrals to mental health clinics and the involvement of fewer providers. This was expected to result in reduced staffing constraints among the already overburdened mental healthcare providers, improved confidentiality, reduced service delays, timely identification and provision of mental health support, and efficiency and convenience of providing services 'under one roof', related to fewer movements and time saved for both AGYW and providers. AGYW and providers also expected integration to result in overall improved quality of care, hence yielding better mental health and HIV outcomes, for instance, reduced CMD symptoms and substance use relapses, improved HIV prevention and treatment adherence and persistence, and viral suppression, as was described by a clinical officer below:

That one [mental health integration] would make your work easier; knowing that all your patients are sane, like they are comfortable, their mental state is okay, it will make your things easier; you will not have to come to a day with a stressed patient who did not take their medication because they were stressed. (Female Clinical Officer, HIV Clinic).

AGYW and providers suggested various strategies to support mental health integration for AGYW within HIV services. These included HIV care providers being more proactive in identifying and addressing mental health issues when providing HIV services, training all HIV care providers, including lay providers, on CMDs and AGYW-friendly communication and counseling skills, and allocating trained mental health specialists like psychologists, nurses, and psychiatrists within HIV clinics. HIV care providers suggested integrating mental health within HIV service protocols to make it easier for them to consider it as part of routine services to AGYW, including providing a defined screening tool to guide screening of AGYW receiving HIV prevention and treatment services for CMDs, reducing daily client bookings in HIV clinics to manage extra workload from mental health services, and creating youth-friendly centers to enhance the privacy and comfort of AGYW receiving HIV and mental health services, as one HIV care nurse suggested below:

The first thing is to have a youth-based center: somewhere they are coming, and they are not lining up, they are not being mixed with those old women; they surely need that privacy. And then those who are seeing them should also be qualified. (Female Nurse, HIV Clinic).

The HIV and mental healthcare providers further recommended the creation of clear linkage and referral systems to ensure better access to mental healthcare for AGYW both within and outside HIV and mental health clinics, including the creation of community-based referral points.

## Discussion

Our findings reveal that AGYW receiving HIV treatment and prevention services experience various CMD symptoms and multi-level risk factors for CMDs but have limited access to mental health services beyond the standard of care HIV and STI counseling. There is a high demand for and acceptability of mental health service delivery within HIV clinics; however, access to these services is notably low among AGYW, and gaps in service delivery exist among providers. Care availability and positive perceptions of mental health service benefits are critical for service utilization among AGYW with CMDs [53]. AGYW described experiences of CMDs using idioms of distress such as 'stress' and 'thinking a lot', which have been commonly described among clients seeking care in Kenya and other SSA settings [54,55]. Key CMD risk factors included HIV stigma among AGYW living with HIV, and financial problems, SRH-related issues, and gender-based violence among AGYW without HIV. These findings are consistent with other studies that have revealed multi-level CMD risk factors

among AGYW, with HIV stigma identified as a key barrier to medication adherence among AGYW living with or at risk of HIV [7,8,15,56,57]. AGYW receiving HIV care in SSA settings have reported CMD symptoms such as suicide ideation linked to individual and interpersonal risk factors, including internalized, enacted, and perceived HIV stigma [58]. Prioritizing routine screening for CMDs and psychosocial interventions that address key CMD risk factors and specific needs of AGYW is essential; and should be integrated in status-neutral HIV service delivery for both AGYW receiving HIV treatment and prevention services [59].

Even though AGYW in this study considered mental health services to be beneficial for addressing their challenges, getting empowerment, and problem-solving, they had limited knowledge of locations where they could access these services. Other studies have shown HIV clinics to be a convenient location for adolescents on HIV care to access mental health support, since these adolescents have reported not knowing alternative locations to access psychosocial support [60]. Providing mental health services to AGYW receiving HIV services could promote their resilience through enhancing their self-efficacy, motivation, and positive attitudes, essential for overcoming risk factors and continued engagement in care, contributing to positive health behaviors and outcomes [61,62]. Previous studies have highlighted the need for psychosocial support to address stigma and disclosure challenges, particularly among AYP living with HIV [58].

In our study, HIV and mental healthcare providers linked AGYW to support groups for mental health, adherence, and substance use issues. Peer support groups aimed at enhancing retention and adherence among AYP are common in Kenyan HIV clinics; however, holistic psychosocial support is often inadequate [63]. Few AGYW in our study had ever received mental health services but perceived the HIV counselling they received as a form of mental health service. This could be due to the individual attention they received from HIV care providers, which they hardly received during routine services. Group-based or individual psychosocial interventions among adolescents living with or affected by HIV in SSA have reported positive emotional and behavioral outcomes [64]. While group-based interventions provide social cohesion and learning, individual models are client-centered, provide individual attention, and address individual needs [65]. Providing individual mental health screening, counseling, and referrals for AGYW within HIV clinics may present an opportunity for client-centered care while addressing CMDs.

We found that AGYW's individual preferences, specifically convenience, interpersonal experiences with providers, and provider demographic characteristics, mainly influenced their decisions to seek mental health services within HIV clinics. This is consistent with studies in SSA and the USA that have identified provider demographic and interpersonal characteristics as valued by youth, and individual-level factors as key barriers to continued adolescent engagement in HIV care [66,67]. Various interpersonal, societal, health system, and policy barriers also hinder mental health service delivery among adolescents in SSA [15,66]. Previous studies have reported that adolescents transitioning to adult HIV services experience anxiety in adjusting to new providers and care settings [13,68]. Non-supportive clinic environments, including stigmatizing attitudes among providers, have also been reported to hinder mental health service delivery to AGYW—for instance, among AGYW with perinatal depression in Nigeria [53]. Our findings indicate that the desire to avoid anxiety, stigma, and judgment from providers likely influenced AGYW's preference for receiving services from familiar, welcoming providers and attending few service delivery points. Sensitivity to individual differences and needs has been emphasized for the successful implementation of mental health interventions among adolescents receiving HIV services [8,29]. Identifying and considering the most relevant preferences and barriers can enhance service access and utilization and improve AGYW's care outcomes [69].

Among providers, individual and structural-level factors, including mental health training, finances, staffing, infrastructure, referral systems, and mental health guidelines, influenced the ability and confidence to deliver mental health services. Lack of training was a key barrier to mental health service delivery among HIV care providers. Shortage of trained mental healthcare providers, lack of regular mental health training, and low confidence and competency to provide mental health services have been described as key barriers in other SSA settings [13,14,16,26]. Inadequate financial and human resource allocations for mental health services have also been highlighted within SSA countries [15]. In our study, some

providers were reluctant to provide services to AGYW due to perceptions that AGYW were 'difficult clients', which hindered mental health service delivery to AGYW. HIV and mental healthcare providers also perceived delivering mental health services to AGYW to be more challenging and time-consuming compared to other populations, and as 'extra work'. A recent study among HIV care providers in Kenya, South Africa, and Zimbabwe reported challenges in delivering PrEP services to adolescent girls younger than 18 years, whom providers described as 'troublesome', 'not listening', and 'dishonest' [70]. Other studies have reported beliefs among providers regarding the perceived difficulty of providing care to adolescents with CMDs [53], and burden from additional screening tools [7]. Studies in SSA have also reported biases among SRH and HIV care providers regarding providing contraception and HIV prevention to AGYW due to cultural stigma towards pre-marital sex among youth [71,72]. In Kenya, low confidence to deliver HIV services to adolescents has been attributed to insufficient provider training on adolescent-friendly HIV care, while access to trained providers and youth-friendly services have been linked to better HIV outcomes [73,74]. In our study, although HIV care providers employed youth-friendly strategies like 'referral by escort', they lacked specific skills to address CMDs and for AGYW-friendly communication and counseling, which likely explains their reluctance to deliver mental health services to AGYW. These findings highlight the need to prioritize regular provider training on mental health service delivery and adolescent-friendly care.

We also found that mental healthcare providers felt more comfortable, competent, and confident in delivering mental health services to AGYW than HIV care providers. HIV care providers' mental health service delivery experiences were largely confined to screening and referral related to HIV treatment non-adherence of AGYW with high viral load, and they preferred referring AGYW with CMDs to trained mental healthcare providers. Our findings are consistent with other studies globally, which have shown that mental health interventions among AGYW largely focus on HIV treatment adherence and SRH outcomes but less on promoting mental health [8]. Current HIV guidelines, such as the Kenya HIV prevention and treatment guidelines (2022) [75], focus on providing mental health screening, referral, and treatment to people living with HIV and HIV treatment clients with detectable viral load, but not other vulnerable populations. However, low screening and treatment for CMDs among populations likely to benefit from PrEP has been shown to result in poorer engagement, adherence, and persistence in HIV prevention care, highlighting the need for prioritization [7,13].

Overall, the integration of mental health within HIV services was highly acceptable among AGYW and providers for addressing CMDs among AGYW receiving HIV services. Integration was expected to potentially yield multiple benefits such as reduced referrals to over-burdened mental healthcare providers, efficiency and convenience of providing services 'under one roof', timely intervention for CMDs, improved quality of care, and better mental health and HIV outcomes. This finding supports other studies that have demonstrated a high demand for mental health integration within HIV services for AGYW [76]. Integrating mental health screening, referrals, and treatment into HIV services, either by multidisciplinary teams providing services 'under one roof' or through coordinated referrals between HIV and mental health clinics, is feasible and acceptable, and potentially addresses AGYW's mental and physical needs [8,26]. Multidisciplinary approaches to screening are strongly suggested to improve access to and uptake of mental health services [13].

Various strategies have been proposed for integrating mental health within HIV services, such as screening for CMDs during PrEP initiation with regular reassessments, and treatment initiation or referrals following CMD diagnosis [7]. Task-shifting strategies involving training and engagement of non-specialist providers in delivering mental health services have also been encouraged in SSA settings to address staffing shortages and increase access to mental health services among adolescents and other vulnerable groups [77,78]. In our study, a proactive approach to mental health service delivery, training of HIV care providers, and integrating mental health within HIV service protocols were suggested as key integration strategies. These could underpin efforts towards the provision of universal mental health screening and integration of mental health treatment into ongoing HIV services [14], and provide evidence for implementing current policies including the Kenya mental health policy (2015–2030), which recognizes the need for integrating mental health within primary healthcare [79]. Future studies could focus on developing and evaluating the feasibility, effectiveness, acceptability, and cost-effectiveness of tailored implementation strategies to support mental health integration for AGYW within HIV

clinics. These strategies could include provider training models on mental health service delivery and AGYW-centered care, integrated mental health service delivery models including within stand-alone youth centers versus HIV clinics and individualized versus group-based psychosocial interventions, specific service integration guidelines, and AGYW-friendly CMD screening tools.

The strengths of this study included enrolling participants actively receiving HIV services and use of the SRQ-20-item mental health screening tool to screen for ongoing mental health symptoms, which ensured that enrolled participants had recent CMD experiences, hence minimizing recall bias and increasing the validity of findings. The study had some limitations. First, we used a small sample size, limiting the generalizability of findings to the general population. Additionally, AGYW had limited experiences with receiving mental health services and were unfamiliar with some of the concepts around integrated service delivery that were discussed, which may have impacted some of their responses about CMD symptoms and mental health services. However, these findings could inform mental health services targeting AGYW in similar settings.

## Conclusion

AGYW receiving HIV services are living with or at substantial risk of HIV infection and experience a high burden of co-occurring CMDs. Our findings highlight critical gaps in integrated mental health and HIV service delivery among AGYW receiving HIV services. The results demonstrate a high demand for and acceptability of mental health integration within HIV clinics but contrastingly low access to the services. Our results show that successful integration would require prioritizing AGYW's unique needs and preferences and consideration of the factors influencing mental health service access and delivery within HIV clinics across all levels of the socio-ecological model. These findings add to the body of evidence supporting the design and implementation of interventions, policies, and guidelines for integrated HIV and mental health service delivery, that are culturally relevant and responsive to AGYW's needs and preferences. Integration and scale-up of mental health services within HIV clinics could improve access to and uptake of mental health services, and improve HIV and mental health outcomes among AGYW in Kenya and similar settings.

## Supporting information

**S1 File. S1 SRQ-20-Item.** SRQ-20-item mental health screening tool.
(PDF)

**S2 File. S2 AGYW interview guide.** AGYW interview guide.
(PDF)

**S3 File. S3 Provider interview guide.** Healthcare provider interview guide.
(PDF)

**S4 File. S4 AGYW codebook.** AGYW codebook.
(PDF)

**S5 File. S5 Provider codebook.** Healthcare provider codebook.
(PDF)

## Acknowledgments

The authors thank the study team, the healthcare providers at PHRD, Thika Sub-County Hospital, all participating HIV and mental health clinics, the AGYW who contributed to this data, and the two reviewers.

## Author contributions

**Conceptualization:** Pamela Kohler, Shannon Dorsey, Nelly Mugo, Jennifer Velloza.

**Formal analysis:** Emmah Owidi, Jennifer Velloza.

**Funding acquisition:** Jennifer Velloza.

**Investigation:** Emmah Owidi, John Njoroge, Snaidah Ayub, Roy Njiru, Boaz Kipkorir, Edinah Casmir, Tessa Concepcion, Tamara Owens.

**Methodology:** Kenneth Ngure, Pamela Kohler, Shannon Dorsey, Jennifer Velloza.

**Project administration:** Kenneth Ngure, Peter Mogere, Catherine Kiptinness, Nelly Mugo.

**Supervision:** Kenneth Ngure.

**Validation:** Bradley H. Wagenaar, Pamela Y. Collins.

**Writing – original draft:** Emmah Owidi.

**Writing – review & editing:** Kenneth Ngure, Peter Mogere, John Njoroge, Snaidah Ayub, Roy Njiru, Boaz Kipkorir, Edinah Casmir, Catherine Kiptinness, Tessa Concepcion, Tamara Owens, Pamela Kohler, Bradley H. Wagenaar, Shannon Dorsey, Pamela Y. Collins, Nelly Mugo, Jennifer Velloza.

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
