## [Decision Letter · Decision Letter 0]

16 Feb 2025

Dear Dr. Velloza,

Thank you for submitting your manuscript to PLOS ONE. After careful consideration, we feel that it has merit but does not fully meet PLOS ONE’s publication criteria as it currently stands. Therefore, we invite you to submit a revised version of the manuscript that addresses the points raised during the review process.

109- 110: explored experiences of mental health service, delivery among AGYW receiving HIV services and healthcare providers in Central, Kenya. Kindly redefine your objectives or research questions?

121: Kindly include subtitles i.e under study population, state/mention the area or site of study, and justifying why you have chosen that particular place, is it most prevalent? or you simply decided, the reason you gave is not justifiable enough to suffice?. Preceded by we recruited 20 AGYW……

121- 122: Clearly explain the description of the sampling strategy, including rationale for the recruitment method, and participant.

Overall, it’s a great article, kindly incorporate the constructive suggestions from the reviewers and myself ie abstracts, methodology and result sections. Thank you

We look forward to receiving your revised manuscript.

Kind regards,

Ajoke Basirat Akinola, Ph.D.

Academic Editor

PLOS ONE

‘The study was funded by a pilot award from the University of Washington Global Mental Health Program to the senior author (JV).”

3. In this instance it seems there may be acceptable restrictions in place that prevent the public sharing of your minimal data. However, in line with our goal of ensuring long-term data availability to all interested researchers, PLOS’ Data Policy states that authors cannot be the sole named individuals responsible for ensuring data access (http://journals.plos.org/plosone/s/data-availability#loc-acceptable-data-sharing-methods).

Additional Editor Comments:

109- 110: explored experiences of mental health service, delivery among AGYW receiving HIV services and healthcare providers in Central, Kenya. Kindly redefine your objectives or research questions?

121: Kindly include subtitles i.e under study population, state/mention the area or site of study, and justifying why you have chosen that particular place, is it most prevalent or you simply decided, the reason you gave is not justifiable to suffice?. Preceded by we recruited 20 AGYW……

121- 122: Clearly explain the description of the sampling strategy, including rationale for the recruitment method, and participant.

Overall, it’s a great article, kindly incorporate the constructive suggestions from the reviewers and myself, thank you.

Abstract:

Overall – the abstract needs work to align with the paper better and have consistent language use, it needs to be rewritten. The SRQ-20 sounds so basic, consider adding that it is a mental health screening tool (or even call it MHST or SRQ-20-MHST). In methods you need to more plainly say that you quantified CMDs (versus “explored experiences of CMDs”) and that you analyzed qualitative data versus just analyzed data using content analysis. You are using a priori coding methods along with inductive methods, you clearly have codes you are interested in (eg., access to services, provider attitudes about mental health services, etc.). What does this mean “AGYW described CMD experiences related to multi-level risk factors…”? This is an example of a rewrite needed. What is a clinician compared to a nurse? A nurse is a clinician too, what cadre are you referring to?

Introduction:

68-69: You state that AGYW were chosen due to disproportionate burden of HIV and CMDs but these data are not shown, what is their burden compared to the general population?

75: There needs to be a better link between these two sentences.

78: Consider adding quick sentence about this paper’s focus.

83: “more than 10%” makes no sense, replace with actual %

85-90: This is too much information about Kenya without anything earlier stating that the study is set there. Consider deleting 87-90 or integrating it elsewhere. This section should be focused first on SSA then go into Kenya (with exception that you use Kenya as an example with references 19-22)

101: “Despite an abundance of research in SSA focused on HIV in adolescents…”. Also the last sentence here “mental health is still poorly understood and under-prioritized within HIV research and interventions…” needs more emphasis. This needs to come through in abstract and at start of introduction, its very important.

109: “experiences with mental health services among AGYW…”. Why Central, Kenya??

Overall: I don’t see GBV mentioned whatsoever and this burden is twinned almost always with HIV for AGYW, this needs to be integrated into literature search and throughout Introduction. An important link is that if AGYW have suffered GBV they are at risk of further harm from GBV (and the associated mental health issues from both) and are at higher risk of HIV, etc.

Methods:

114: This first sentence needs to be rewritten, what is the purpose of the JiTunze study?

Study Population section:

This is a bit of a mish mash. The sampling methods should be its own section. The whole section needs a rewrite to be more plain and simple.

122: What kind of interviews?

123: screening and enrollment for the study?

124: PHRD, add if this is a clinic, a hospital, what kind of facility? Tertiary? We can tell what a subcounty hospital may be like but not PHRD.

127: See my notes about renaming SRQ-20 to be less vague.

131: “their capacity to consent”. I’m glad you had this procedure in the study. Do you know how many referrals from the study were actually linked to services?

138: isn’t the sample of providers only coming from within the 2 identified facilities, not any facility as you state?

141: “all interviews with AGYW and providers…” and (author EO)

155: Did you stop the interview or complete later? Did they go for support after the interview was completed?

157: We directly transcribed audio-recorded interviews verbatim regardless of language and translated Kiswahili transcripts into English.

175: consider “multilevel factors” instead of social factors (esp. since that word shows up later in the paper) and also including why this model is good for AGYW (very much affected by circles of influence outside of themselves) and add more literature for why this model fits well for AGYW.

Ethical statement: what did you do with the audio recordings, including phone recordings.

Results

Participant characteristics: is there a reason you aren’t including # HIV negative versus # HIV positive? This is relevant I think to the findings presented in the paper. I would add in addition to the number of providers and their role, how many from each facility? And what kind of facility?

AGYW experience of CMD is one section and “Risk factors for CMDs” is another.

Overall: sometimes you say experiences with CMD and sometimes symptoms, which one is best to use and stick with it. You definitely have a priori themes in the results, you need to change methods to say you had a priori codes you were looking for in the data.

215: the fact that you are wanting to find individual level risk factors is not coming through in abstract, introduction or methods.

216: “risk factor for experiencing…”

219: “and fear of potentially transmitting…”

223: Maybe just me but I would want to state the service, like “as one AGYW seeking PrEP reported:”

244: This is very interesting that AGYW are conflating any individual attention given to somehow receiving counseling…this is not commented on in discussion and I think it should be. For one, a simple change in practice is to have more individual attention and less group adherence counseling which occurs in a lot of places. Did you ask more about group versus individual services? I would look for any qualitative data about group versus individual counseling for AGYW getting HIV services and see if you see anything interesting that can be added to the paper. For sure I think group counseling may be an opportunity lost to talk to an individual AGYW about stress, anxiety, depression, GBV, poverty, etc. Generally, this is an excellent section…

261: Interesting that this quote is essentially the definition of resiliency. Maybe comment on that here or in discussion? Also anything to add to discussion about stress? It seems a common complaint but also vague, any literature showing that AGYW or others use this word as an all encompassing word for a mental health concern? I heard it a lot in these kinds of settings too.

273: “delivering mental health services to AGYW than HIV providers did”. “However, HIV care providers’ lower confidence in providing mental health services was mainly due to…”

277: “however, more specific screening tools …”

279: internalized stigma

285: I have never heard this ever that an HIV clinic had support groups for AGYW unless it was an outside program like DREAMS, it just doesn’t happen. I would verify this information with Kenyan colleagues or make a note anywhere you talk about support groups that they were non-governmental additional services brought by an outside group…unless he is totally making this up to make the clinic look good?

297: You just wrote that AGYW get referred. So is this counseling done by the clinic where the AGYW presented or at the referral site? Need to clarify.

306: This is another quote that needs verification I believe. You may just want to find out what % of AGYW (or any patient) is physically escorted. This is so uncommon in these settings and this nurse may not be entirely truthful about all AGYW being physically escorted, this is so uncommon in a very busy clinic that staff can escort patients. Or explain if we are talking about a different department at a hospital??

310: Consider ‘access’ instead of delivery and ‘by AGYW’; check abstract an intro to see if factors influencing access to services is coming through strongly enough …

339: consider, “Conversely, mental healthcare providers…”

344: rewrite for better clarity

349: Don’t need however. What does “difficulty of AGYW gaining insight” mean?

350: mental health stigma?

353-359: I don’t see this quote as showing that some people think mental health care providers behave like their clients, I would delete that conclusion. They are just saying that providers may be afraid of MH patients because they could be aggressive or harm them.

366: “since adolescents had more routine HIV care parameters” what does this mean? Rewrite these sentences here because it is important to note clearly why providers thought it was different to work with adolescents and why exactly.

369: “screening into HIV clinics” or into HIV service delivery

389: Benefit of integrating …?

390-396: needs rewrite for more clarity

409: CMD symptoms, substance use relapses, …

415-431: See notes above about individual versus group counseling, did this come up in the interviews re. how to integrate MH services

Discussion

436: “contrasting service access and delivery gaps among AGYW and providers”. What does this mean? Can be stated in plain language like differences of opinion or something like that.

446-450: This jumps out early in discussion, esp. after just reading those findings the page before. Suggest moving this to join with other service integration language later in Discussion.

473-476: why are they more challenging? I’m still not seeing why. Cite any literature where providers consider serving adolescents as like serving children, they are looked down on as a population to serve.

Overall: it seems that you have chosen to endorse integration over a stand alone youth center, as mentioned by the providers and which is in fact is a very common way to deal with the very real difficulties in integrating adolescent care into an adult setting. This again is a priori coding and should be stated as such, that you were interested in finding out AGYW and providers view on integration…but take a look to see if you can also mention that some providers support youth centers as an alternative and that perhaps future studies should compare and contrast the two models for adolescent mental health, including cost, to determine which model leads to better outcomes.

Please see attached Word document with detailed review of paper. Overall: it seems that you have chosen to endorse integration over a stand alone youth center, as mentioned by the providers and which is in fact is a very common way to deal with the very real difficulties in integrating adolescent care into an adult setting. This again is a priori coding and should be stated as such, that you were interested in finding out AGYW and providers view on integration…but take a look to see if you can also mention that some providers support youth centers as an alternative and that perhaps future studies should compare and contrast the two models for adolescent mental health, including cost, to determine which model leads to better outcomes.

Abstract

- Please add to methods what the time range for data collection was

- In results, please clarify if systematic or systemic lack of access to MH services was meant by authors.

- Results seem to switch back and forth between AGYW and providers. Please reorganize and/or clarify how socioecological model informs the structure of the results.

- Results around providers feeling less trained compared to mental health providers is unclear that mental health providers were included in the sample of interviews.

Introduction

Overall, excellent introduction to the availability of MH services in Kenya and the overlaps between HIV and CMDs.

- Line 92 unclear if “inadequate mental healthcare providers” is referring to availability of providers or the training is inadequate. Please clarify

Methods

Thorough explanation of the methodology employed for this study, and gives a good sense of what was covered in the interviews and why those were of interest based on the introduction.

- Line 121 – what kind of providers are included in this category? Please add more detail as to the educational background to qualify for interviews.

Results

Quite a lot of themes to cover in this section, but overall great work in explaining the depth of information available in these interviews.

- Please add a summary of themes to better orient readers to the framework and organization of the results.

- Line 197 unclear how AGYW experiences and risk factors for CMD were a separate theme from Individual-level risk factors. Or whether they were an introduction to the risk factors. Please add language to delineate.

Discussion

Well-written contextualization of their data within existing knowledge base.

- Line 455, These results really stands out as a reader. Are these biases against working with adolescents unique to this subset of providers or are there other studies that reflect this reluctance?

Reviewers' comments:

Reviewer's Responses to Questions

**Comments to the Author**

1. Is the manuscript technically sound, and do the data support the conclusions?

Reviewer #1: Yes

Reviewer #2: Yes

2. Has the statistical analysis been performed appropriately and rigorously?

Reviewer #1: Yes

Reviewer #2: Yes

3. Have the authors made all data underlying the findings in their manuscript fully available?

Reviewer #1: Yes

Reviewer #2: Yes

4. Is the manuscript presented in an intelligible fashion and written in standard English?

Reviewer #1: Yes

Reviewer #2: Yes

Reviewer #1: Abstract

- Please add to methods what the time range for data collection was

- In results, please clarify if systematic or systemic lack of access to MH services was meant by authors.

- Results seem to switch back and forth between AGYW and providers. Please reorganize and/or clarify how socioecological model informs the structure of the results.

- Results around providers feeling less trained compared to mental health providers is unclear that mental health providers were included in the sample of interviews.

Introduction

Overall, excellent introduction to the availability of MH services in Kenya and the overlaps between HIV and CMDs.

- Line 92 unclear if “inadequate mental healthcare providers” is referring to availability of providers or the training is inadequate. Please clarify

Methods

Thorough explanation of the methodology employed for this study, and gives a good sense of what was covered in the interviews and why those were of interest based on the introduction.

- Line 121 – what kind of providers are included in this category? Please add more detail as to the educational background to qualify for interviews.

Results

Quite a lot of themes to cover in this section, but overall great work in explaining the depth of information available in these interviews.

- Please add a summary of themes to better orient readers to the framework and organization of the results.

- Line 197 unclear how AGYW experiences and risk factors for CMD were a separate theme from Individual-level risk factors. Or whether they were an introduction to the risk factors. Please add language to delineate.

Discussion

Well-written contextualization of their data within existing knowledge base.

- Line 455, These results really stands out as a reader. Are these biases against working with adolescents unique to this subset of providers or are there other studies that reflect this reluctance?

Reviewer #2: Please see attached Word document with detailed review of paper. Overall: it seems that you have chosen to endorse integration over a stand alone youth center, as mentioned by the providers and which is in fact is a very common way to deal with the very real difficulties in integrating adolescent care into an adult setting. This again is a priori coding and should be stated as such, that you were interested in finding out AGYW and providers view on integration…but take a look to see if you can also mention that some providers support youth centers as an alternative and that perhaps future studies should compare and contrast the two models for adolescent mental health, including cost, to determine which model leads to better outcomes.

**Do you want your identity to be public for this peer review?** For information about this choice, including consent withdrawal, please see our Privacy Policy

Reviewer #1: No

Reviewer #2: **Yes: ** Ellen W. Maclachlan

---

## [Author Response · Author response to Decision Letter 1]

29 Mar 2025

18th March 2025

The Academic Editor

PLOS ONE

Dear Dr., Akinola,

RE: Response to Manuscript ID PONE-D-24-60333

We thank the reviewers for your review of our manuscript entitled, “Mental health service delivery among adolescent girls and young women (AGYW) seeking HIV prevention and treatment services in central Kenya: a qualitative study of AGYW and healthcare providers’ perceptions”, and the opportunity to resubmit a revised version. We appreciate your and the reviewers’ valuable feedback, and we are pleased to incorporate these recommendations. We are resubmitting the revised manuscript that addresses all suggestions. Below is an updated point-by-point response to the comments along with a description of the changes that have been made to address the comments. The reviewer comments are included in bold text, and our responses are standard font. We appreciate, in advance, your time and consideration of this revised manuscript for publication in PLOS ONE.

Sincerely,

Jennifer Velloza (on behalf of all authors)

University of California San Francisco (UCSF)

Editor Comments:

We have made changes throughout the manuscript to meet PLOS ONE’s style requirements, including file naming, layout, headings, citation formatting, and supporting information.

‘The study was funded by a pilot award from the University of Washington Global Mental Health Program to the senior author (JV).”

We have included this funding statement in the cover letter, which now reads: “This study was funded by a pilot award from the University of Washington Global Mental Health Program to the senior author (JV). The funders had no role in the study design, data collection and analysis, decision to publish, or preparation of the manuscript.”

3. In this instance it seems there may be acceptable restrictions in place that prevent the public sharing of your minimal data. However, in line with our goal of ensuring long-term data availability to all interested researchers, PLOS’ Data Policy states that authors cannot be the sole named individuals responsible for ensuring data access (http://journals.plos.org/plosone/s/data-availability#loc-acceptable-data-sharing-methods).

We have updated the data availability statement and included an institutional contact: the International Clinical Research Center (ICRC) at the University of Washington. Staff at the ICRC were not directly involved in the study and are not listed as co-authors on the manuscript. They, however, have access to the data and can respond to external requests for data. Lines 734-741 now read: “The semi-structured interview guides and codebooks used to collect and analyze the qualitative interviews are included as additional files. De-identified transcripts may be available after consultation with the University of Washington and Kenya Medical Research Institute institutional review boards, as per the ethical approval obtained for this study. The full de-identified transcripts will be available upon author request and under appropriate data-sharing agreements to those who provide a methodologically sound proposal by contacting the International Clinical Research Center at the University of Washington (Email: icrc@uw.edu) or the senior author (Email: jennifer.velloza@ucsf.edu).”

The interview transcripts contain potentially identifying and sensitive information related to young women’s (including minors’) experiences of HIV risk factors and mental health symptoms that could potentially be harmful if their identity was disclosed. Considerable privacy concerns have been noted regarding the confidentiality of qualitative interviews done with adolescents living with HIV and those engaged in high-risk sexual behaviour (MacDonald et al., 2023: 10.1016/j.jpeds.2023.113589). Other researchers (e.g., Chauvette et al., 2019: https://doi.org/10.1177/1609406918823; and Lamb et al., 2024: https://doi.org/10.1177/13591053241237) have also highlighted various epistemological, methodological, legal, and ethical concerns regarding open data sharing and secondary analysis of qualitative research. These concerns include privacy and confidentiality and potential harm to participants whose identity may be easily reconstructed, especially in small rural communities such as among the adolescents and young women in our peri-urban study setting.

We have revised the data availability statement in lines 734-741 which now reads: “The semi-structured interview guides and codebooks used to collect and analyze the qualitative interviews are included as additional files. De-identified transcripts may be available after consultation with the University of Washington and Kenya Medical Research Institute institutional review boards, as per the ethical approvals obtained for this study. The full de-identified transcripts will be available upon author request and under appropriate data-sharing agreements to those who provide a methodologically sound proposal by contacting the International Clinical Research Center at the University of Washington (Email: icrc@uw.edu) or the senior author (Email: jennifer.velloza@ucsf.edu).”

Additional Editor Comments:

109- 110: explored experiences of mental health service, delivery among AGYW receiving HIV services and healthcare providers in Central, Kenya. Kindly redefine your objectives or research questions?

We have made changes to redefine the objectives in the introduction section. Lines 142-144 now read: “We conducted a qualitative study to understand the perceptions of mental health service access and delivery within HIV clinics among AGYW receiving HIV services and healthcare providers in Central, Kenya.”

121: Kindly include subtitles i.e under study population, state/mention the area or site of study, and justifying why you have chosen that particular place, is it most prevalent or you simply decided, the reason you gave is not justifiable to suffice?. Preceded by we recruited 20 AGYW……

We have revised the sub-headings to separate the study design and the study setting and population in the methods section. Lines 157-168 now read: “We conducted the study in Thika, an urban center in Kiambu county in Central Kenya, located approximately 40km from the capital, Nairobi, with a population-level HIV prevalence of 2.7% and a large surrounding peri-urban and rural population [36]. Kiambu county hosts approximately 250,000 AGYW of ages 15-24, who account for over 1,000 new HIV infections annually [37]. The primary site for the study, Thika sub-County Hospital (Level 5), was ideal for the study as it provides HIV prevention and treatment services to a large population of AGYW aged 10-24 (�233 living with HIV; and 9 on PrEP). We also included seven surrounding healthcare facilities to increase the diversity of the study population. These included Partners in Health and Research Development (PHRD), a clinical research site that implements HIV prevention and treatment trials among diverse populations including AGYW, Makongeni dispensary (Level 2), Kiandutu health centre (Level 3), Ruiru sub-County hospital (Level 4), Kiambu County referral hospital (Level 5), Mathari National referral hospital (Level 6), and Kenyatta National hospital (Level 6).”

121- 122: Clearly explain the description of the sampling strategy, including rationale for the recruitment method, and participant.

We have clarified and included the rationale for the sampling and recruitment strategy. Lines 171-181 now read: “AGYW were referred for screening and enrollment for the study by HIV care providers from Thika sub-County Hospital and the PHRD clinic. Eligible AGYW were aged 16-25 years and were currently receiving HIV prevention, treatment, or testing services at either clinic at the time of enrollment. HIV care providers at the two facilities were sensitized about the study to support the recruitment of AGYW by identifying potential participants while providing HIV services. Once the providers identified AGYW who fit the eligibility criteria, they informed them about the study and referred those who expressed willingness for study participation to the study staff at the PHRD clinic (approximately 1.5 kilometers distance) for screening and enrolment. This assured AGYW of confidentiality, enhanced their comfort in study participation, and built trust between the AGYW and study staff.”

Lines 198-209 now read: “We used stratified purposive sampling to include AGYW in different age groups (16-18, 19-22, and 23-25); and receiving different HIV prevention and treatment services: PrEP, HIV testing services (HTS), post-exposure prophylaxis (PEP), and antiretroviral therapy (ART), to capture variations in experiences of mental health service delivery among different demographic groups [40]. We also purposively sampled HIV and mental healthcare providers to include different cadres (HIV counsellors, clinical officers, nurses, psychiatrists, and psychologists), healthcare facility representation, service delivery experience, and educational backgrounds (college diploma and university degree). The overall sample size of 20 AGYW and 10 providers was determined by thematic saturation, whereby we stopped interviews once no new data emerged [40].”

Overall, it’s a great article, kindly incorporate the constructive suggestions from the reviewers and myself, thank you.

Thank you for this positive feedback.

Reviewer 1 comments

Reviewer #1: Abstract

- Please add to methods what the time range for data collection was

We have included the timeframe in the abstract. Lines 40-42 now read: “Between 16th February and 14th June 2021, we conducted in-depth interviews with AGYW receiving HIV services and healthcare providers from eight clinics in Central Kenya.”

- In results, please clarify if systematic or systemic lack of access to MH services was meant by authors.

We have revised the sentence in lines 56-57, which now reads: “AGYW reported a high need for mental health services but described a systemic lack of access.”

- Results seem to switch back and forth between AGYW and providers. Please reorganize and/or clarify how socioecological model informs the structure of the results.

We have reorganized the results in the abstract, and lines 54-64 now read: “AGYW described experiences of CMDs attributed to individual, interpersonal, and community-level risk factors including HIV stigma, financial, and relationship challenges. AGYW reported a high need for mental health services but described a systemic lack of access. Convenience and positive experiences with providers facilitated AGYW’s access to services. Conversely, HIV care providers felt less confident in delivering mental health services due to inadequate mental health training compared to mental healthcare providers. Providers also reported inadequate training, poor referral systems, and unclear guidelines that hindered service delivery. AGYW and providers endorsed mental health service integration within HIV clinics to potentially reduce referral burden for AGYW and improve service quality.”

- Results around providers feeling less trained compared to mental health providers is unclear that mental health providers were included in the sample of interviews.

We have clarified the methods in the abstract to include mental healthcare providers. Lines 51-53 now read: “The providers (n=10) comprised seven females, and included six HIV and four mental healthcare providers.”

Introduction

Overall, excellent introduction to the availability of MH services in Kenya and the overlaps between HIV and CMDs.

Thank you for this positive feedback.

- Line 92 unclear if “inadequate mental healthcare providers” is referring to availability of providers or the training is inadequate. Please clarify

We have revised the sentence in lines 111-114 to now read: “Numerous barriers hinder mental health service delivery among adolescents receiving HIV services in SSA, including stigma, shortage of mental healthcare providers, poor treatment and referral pathways, and a lack of routine screening for CMDs [13,15,16].”

Methods

Thorough explanation of the methodology employed for this study, and gives a good sense of what was covered in the interviews and why those were of interest based on the introduction.

Thank you for this positive feedback.

- Line 121 – what kind of providers are included in this category? Please add more detail as to the educational background to qualify for interviews.

We have revised the methods section to include the provider cadres and educational background. Lines 202-205 now read: “We also purposively sampled HIV and mental healthcare providers to include different cadres (HIV counsellors, clinical officers, nurses, psychiatrists, and

---

## [Decision Letter · Decision Letter 1]

1 Jul 2025

Dear Dr. Velloza,

We look forward to receiving your revised manuscript.

Kind regards,

Tai-Heng Chen, M.D., Ph.D.

Academic Editor

PLOS ONE

Reviewers' comments:

Reviewer's Responses to Questions

**Comments to the Author**

Reviewer #1: All comments have been addressed

Reviewer #2: All comments have been addressed

2. Is the manuscript technically sound, and do the data support the conclusions?

Reviewer #1: Yes

Reviewer #2: Partly

3. Has the statistical analysis been performed appropriately and rigorously?

Reviewer #1: N/A

Reviewer #2: Yes

4. Have the authors made all data underlying the findings in their manuscript fully available?

Reviewer #1: Yes

Reviewer #2: Yes

5. Is the manuscript presented in an intelligible fashion and written in standard English?

Reviewer #1: Yes

Reviewer #2: Yes

Reviewer #1: Thank you for the thorough responses to my reviewer suggestions. The manuscript reads clearly and concisely, and will be of great value to the field.

Reviewer #2: A much better draft, well done. Please see my notes for a final revision. This is much improved from the original. In the methods I would say more about the items, scales or questions in the SRQ-20. A major limitation is that the MH services in these settings largely don’t exist so the interviews are partly based on a hypothetical, but this is not addressed in discussion. Also, the discussion doesn’t really address whether HIV positive or HIV negative (seeking prevention services) need different MH services or perhaps if HIV positive AGYW should at least at the beginning, be prioritized.

**Do you want your identity to be public for this peer review?** For information about this choice, including consent withdrawal, please see our Privacy Policy

Reviewer #1: No

Reviewer #2: **Yes: ** Ellen W. Maclachlan

---

## [Author Response · Author response to Decision Letter 2]

14 Aug 2025

14th August 2025

The Academic Editor

PLOS ONE

Dear Dr. Chen,

RE: Response to Manuscript ID PONE-D-24-60333R1

We thank the reviewers for your review of our manuscript entitled, “Mental health service delivery among adolescent girls and young women (AGYW) seeking HIV prevention and treatment services in central Kenya: a qualitative study of AGYW and healthcare providers’ perceptions”, and the opportunity to resubmit a revised version. We appreciate your and the reviewers’ valuable feedback, and we are pleased to incorporate these recommendations. We are resubmitting the revised manuscript that addresses all suggestions. We noticed that we had already addressed some of the comments in the previous revision submitted on 30th March 2025 (Manuscript Number: PONE-D-24-60333R1), which we have indicated in our responses here. Below is an updated point-by-point response to the comments along with a description of the changes that have been made to address the comments. Page and line numbers indicated below refer to the clean version of the manuscript. The reviewers’ comments are included in bold text, and our responses are standard font. We appreciate, in advance, your time and consideration of this revised manuscript for publication in PLOS ONE.

Sincerely,

Jennifer Velloza, PhD, MPH (on behalf of all authors)

University of California, San Francisco (UCSF)

Reviewer #1 Comments:

Thank you for the thorough responses to my reviewer suggestions. The manuscript reads clearly and concisely, and will be of great value to the field.

We thank the reviewer for this positive feedback.

Reviewer #2 Comments:

A much better draft, well done. Please see my notes for a final revision. Overall: This is much improved from the original.

We thank the reviewer for this positive feedback.

In the methods I would say more about the items, scales or questions in the SRQ-20.

For this study, our analysis focused on the qualitative findings since the SRQ-20-item screening tool was only used for eligibility assessment for interview participation. We provided more detail on the SRQ-20 in our primary quantitative paper: Concepcion et al., (2023). Higher rates of mental health screening of adolescents recorded after provider training using simulated patients in a Kenyan HIV clinic: results of a pilot study. doi: 10.3389/fpubh.2023.1209525, which we have cited in line 128 of the current manuscript. We have also included a copy of the SRQ-20-item questionnaire that we used for screening AGYW as a supporting file which is cited in line 160 for further reference.

A major limitation is that the MH services in these settings largely don’t exist, so the interviews are partly based on a hypothetical, but this is not addressed in discussion.

We have included a discussion on AGYW’s limited experiences with receiving mental health services in the discussion section, and lines 653-657 now read: “The study had some limitations. First, we used a small sample size, limiting the generalizability of findings to the general population. Additionally, AGYW had limited experiences with receiving mental health services and were unfamiliar with some of the concepts around integrated service delivery that were discussed, which may have impacted some of their responses about CMD symptoms and mental health services.”

Also, the discussion doesn’t really address whether HIV positive or HIV negative (seeking prevention services) need different MH services or perhaps if HIV positive AGYW should at least at the beginning, be prioritized.

From our findings, mental health services are important and should be prioritized for all AGYW receiving HIV services regardless of their HIV status. We have included a discussion on the prioritization of MH services for AGYW in lines 541-543 which now read: “Prioritizing routine screening for CMDs and psychosocial interventions that address key CMD risk factors and specific needs of AGYW is essential; and should be integrated in status-neutral HIV service delivery for both AGYW receiving HIV treatment and prevention services [59].”

Abstract:

34: consider AGYW instead of ‘they’

We previously addressed this comment and made changes to line 34-35 which now reads: “However, mental health is under-prioritized within HIV interventions targeting AGYW.”

39: do you need apostrophe at the end?

We previously addressed this comment and removed the apostrophe in lines 38-39 which now read: “Between 16th February and 14th June 2021, we conducted in-depth interviews with AGYW receiving HIV services and healthcare providers from eight clinics in Central Kenya.”

43: consider revising to make clear this is AGYW only; perhaps revise to include providers somehow

We have revised the sentence to specify experiences of CMDs among AGYW and included providers, and lines 42-44 now read: “Interviews explored AGYW’s experiences with CMDs and factors influencing mental health service delivery by providers within HIV clinics.”

50: consider “due” multi-level risk factors and also “financial problems”

We have revised the sentence and lines 49-50 now read: “AGYW described experiences of CMDs due to multi-level risk factors, including HIV stigma, financial problems, and relationship challenges.”

51: you refer to need throughout, this could be considered “demand” for mental health services

We have revised this sentence and throughout the manuscript and lines 50-51 now read: “AGYW reported a high demand for mental health services but described a systemic lack of access.”

Introduction:

73: “poor adherence” leading to uncontrolled viral load and poor treatment outcomes (I believe this should be spelled out more since it is a serious consequence)

We have made the suggested changes to lines 81-84 which now read: “CMDs undermine HIV outcomes among AGYW by contributing to low service uptake and retention, poor adherence leading to uncontrolled viral load and poor treatment outcomes, limited engagement and persistence in care, and increased sexual risk behaviors, exacerbating AGYW’s risk of HIV acquisition and transmission.”

91-93: consider moving to earlier in introduction (line 82)

We have made changes and moved this sentence to earlier in the introduction in lines 77-80 which now read: “Numerous barriers hinder mental health service access and delivery among adolescents receiving HIV services in SSA, including stigma, shortage of trained mental healthcare providers, poor treatment and referral pathways, and a lack of routine screening for CMDs [13–15].”

108-109: “explored mental health service delivery” this is partially misleading the reader, since you clearly state that such services don’t exist per se beyond adherence counseling; you are more exploring the potential that exists

We have made changes to the sentence in lines 117-119 which now read: “We conducted a qualitative study to understand the perceptions around potential expansion of mental health service access and delivery within HIV clinics among AGYW receiving HIV services and healthcare providers in Central Kenya.”

110: take out apostrophe after Central

We have made changes to the sentence in lines 117-119 which now read: “We conducted a qualitative study to understand the perceptions around potential expansion of mental health service access and delivery within HIV clinics among AGYW receiving HIV services and healthcare providers in Central Kenya.”

Methods:

Even though you don’t present findings on the SRQ-20 (you may want to consider including these data to make the paper more robust, though I realize it is a qualitative report) it would be good to get some inkling of the main domains in this tool, what is measured, etc.

For this study, our analysis focused on the qualitative findings since the SRQ-20-item screening tool was only used for eligibility assessment for interview participation. We provided more detail on the SRQ-20 in our primary quantitative paper: Concepcion et al., (2023). Higher rates of mental health screening of adolescents recorded after provider training using simulated patients in a Kenyan HIV clinic: results of a pilot study. doi: 10.3389/fpubh.2023.1209525, which we have cited in line 128 of the current manuscript. We have also included a copy of the SRQ-20-item questionnaire that we used for screening AGYW as a supporting file which is cited in line 160 for further reference.

122: add “study-related” screening and enrollment, I read it like they getting services

We have made this change and lines 149-150 now read: “AGYW were referred for study-related screening and enrollment for the study by HIV care providers from Thika sub-County Hospital and the PHRD clinic.”

126: where did screening occur? In the clinic? Was it private?

Screening occurred in a private clinic room at the PHRD clinic. We have made this change, and lines 158-161 now read: “Once the AGYW arrived at the PHRD clinic, they were screened for study participation by a study psychosocial counselor in a private clinic room using the Self Reporting Questionnaire 20-item (SRQ-20) mental health screening tool [38] (Supporting File 1), to assess for elevated symptoms of CMDs.”

129: eligible for in depth interviews

We have made this change and lines 161-162 now read: “AGYW with mild-to-moderate CMD symptoms (determined by a score of 7-14 on the SRQ-20 screening tool), were eligible for in-depth interviews.”

131: consent to participation

We have made this change and lines 162-165 now read: “AGYW with scores above 14, indicating severe CMD symptoms, were excluded from interviews due to potential impacts on the capacity to consent to study participation, and risk of harm related to emotional distress from discussing sensitive topics or traumatic experiences [39].”

132: start new sentence after the age groups are noted

We have made this change to start a new sentence, and lines 174-177 now read: “We included AGYW in different age groups (16-18, 19-22, and 23-25). These AGYW were receiving a variety of HIV prevention and treatment services including: PrEP, HIV testing services (HTS), post-exposure prophylaxis (PEP), and antiretroviral therapy (ART).”

136: consider new paragraph for methods related to providers

We have started a new paragraph for methods related to providers in lines 167-171 and lines 178-182.

138: this is a hold over from previous draft; I am still confused that you state earlier you are working in 2 clinics but then the providers are from a much larger region; perhaps you can add a statement that providers were not just recruited from the two clinics where the AGYW were seeking HIV services but from a broader area (and why you recruited that way)

We have made changes to clarify the study recruitment sites for AGYW and healthcare providers at the beginning of the participant recruitment section and lines 144-148 now read: “AGYW were recruited from Thika sub-County Hospital and the PHRD clinic. Healthcare providers were recruited from Thika sub-County Hospital and six surrounding facilities to capture broader perspectives on mental health service delivery to AGYW within the larger catchment area that provides referrals to the primary study site.”

We also included the rationale for involving additional facilities earlier in the study setting and population section in lines 136-141 which now read: “We also included seven surrounding healthcare facilities to increase the diversity of the study population. These included Partners in Health and Research Development (PHRD), a clinical research site that implements HIV prevention and treatment trials among diverse populations including AGYW, Makongeni dispensary (Level 2), Kiandutu health centre (Level 3), Ruiru sub-County hospital (Level 4), Kiambu County referral hospital (Level 5), Mathari National referral hospital (Level 6), and Kenyatta National hospital (Level 6).”

Data collection explanation is very good

We thank the reviewer for this positive feedback.

169: it isn’t clear if the social-ecological model was used for both sets of IDIs? For the providers as well?

Yes, we used the socio-ecological model for both sets of IDIs. We have clarified this in the data management section and lines 219-222 now read: “We used the socio-ecological model to categorize themes around the individual, interpersonal, community, and structural factors influencing mental health service delivery, access, and utilization among AGYW receiving HIV services and providers delivering HIV and mental health services in the respective facilities.”

Results:

196: add range for years of service for providers

We have made the suggested change and lines 258-261 now read: “Providers had a median age of 37 years (IQR: 30-44). The sample included 70% (n=7) females, with a median of 7 years of experience (IQR: 5-17) in HIV or mental health service delivery; their educational backgrounds included diploma (n=6), higher diploma (n=2) and degree (n=2).”

198: consider replacing “broadly” with often

We have made this change and lines 266-268 now read: “AGYW often described any experiences of CMDs as ‘stress’ and used words in English, Kiswahili, or ‘Sheng’ (an urban slang popular among Kenyan youth) languages to describe their mental health symptoms.”

206: “were more likely”

We have made this change and lines 273-275 now read: “Symptoms like self-isolation, changes in sleep and appetite, and drug and alcohol use were more likely to be reported among AGYW with lower SRQ-20 screening scores (7-10).”

216: “for experiencing CMDs”

We previously addressed this comment and made changes to line 285 which now reads: “Among AGYW living with HIV, HIV stigma was a common risk factor for experiencing CMDs.”

223: remove apostrophe after condom use

We previously addressed this comment and removed the apostrophe in lines 291-293 which now read: “Other issues included infertility, sexually transmitted infections (STIs), and challenges negotiating condom use and HIV testing with sexual partners, as one AGYW seeking PrEP reported below:”

228: this would be more clear to me if you talk about AGYW experiences of mistreatment due to HIV-related stigma (or even HIV status related…this does not get explicitly stated in the paper, it is assumed we know the stigma is due to HIV status)

We have clarified the results to specify HIV-related stigma as a risk factor for CMDs. Lines 298-303 now read: “Among AGYW living with HIV, CMD symptoms were also attributed to experiences of HIV-related stigma from discrimination, abuse, and withholding of emotional, material, and treatment support by peers, partners, and caregivers due to AGYW’s HIV status. Furthermore, perceived and anticipated HIV-related stigma was related to lack of privacy in HIV clinics, schools, and the community, which presented a risk of inadvertent HIV status disclosure and affected discreetness in taking medication.”

245: the fact that less than half of the AGYW reported receiving mental health services is a limitation of this study; they do not have the experience from which to draw their opinions and ideas, which is central to qualitative work, add to limitations

We have included a discussion on AGYW’s limited experiences with receiving mental health services in the discussion section, and lines 653-657 now read: “The study had some limitations. First, we used a small sample size, limiting the generalizability of findings to the general population. Additionally, AGYW had limited experiences with receiving mental health services and were unfamiliar with some of the concepts around integrated service delivery that were discussed, which may have impacted some of their responses about CMD symptoms and mental health services.”

248-255: this whole section is making conclusions with no quote, perhaps add a quote here

We have included a quote to support the results as suggested. Lines 319-328 now read: “However, we found that most AGYW who reported receiving mental health services in HIV clinics had only received HIV adherence counseling and follow-up specific to HIV and STI prevention and treatment services (e.g., HIV testing, HIV status disclosure, and medication adherence) from HIV counseling providers, as one AGYW seeking PEP described below:

Okay, what she [provider] did is the counseling: she told me about how I c

---

## [Decision Letter · Decision Letter 2]

14 Nov 2025

Mental health service delivery among adolescent girls and young women (AGYW) seeking HIV prevention and treatment services in central Kenya: a qualitative study of AGYW and healthcare providers’ perceptions

PONE-D-24-60333R2

Dear Dr. Velloza,

We’re pleased to inform you that your manuscript has been judged scientifically suitable for publication and will be formally accepted for publication once it meets all outstanding technical requirements.

Kind regards,

Hong-Van Tieu

Academic Editor

PLOS ONE

Additional Editor Comments (optional):

Thank you for your careful consideration of the reviewers' comments and incorporating into the revised manuscript.

Reviewers' comments:

Reviewer's Responses to Questions

**Comments to the Author**

Reviewer #1: All comments have been addressed

2. Is the manuscript technically sound, and do the data support the conclusions?

Reviewer #1: Yes

3. Has the statistical analysis been performed appropriately and rigorously?

Reviewer #1: Yes

4. Have the authors made all data underlying the findings in their manuscript fully available?

Reviewer #1: Yes

5. Is the manuscript presented in an intelligible fashion and written in standard English?

Reviewer #1: Yes

Reviewer #1: No new comments. Authors have successfully presented their findings in a concise manner and contextualized them thoroughly in the literature.

**Do you want your identity to be public for this peer review?** For information about this choice, including consent withdrawal, please see our Privacy Policy

Reviewer #1: No

---

## [Editor Report · Acceptance letter]

PONE-D-24-60333R2

PLOS ONE

Dear Dr. Velloza,

I'm pleased to inform you that your manuscript has been deemed suitable for publication in PLOS ONE. Congratulations! Your manuscript is now being handed over to our production team.

Kind regards,

on behalf of

Dr. Hong-Van Tieu

Academic Editor

PLOS ONE